

# Catch composition and risk assessment of two fishing gears used in small-scale fisheries of Bandon Bay, the Gulf of Thailand

Tuantong Jutagate[1] and Amonsak Sawusdee[2]

[1] Faculty of Agriculture, Ubon Ratchathani University, Warin Chamrab, Ubon Ratchathani, Thailand
[2] School of Science, Walailak University, Thasala, Nakhon Si Thammarat, Thailand

Corresponding author
Amonsak Sawusdee,
samonsak@wu.ac.th

## ABSTRACT

We examined catch compositions and vulnerability of target and bycatch species in two fishing gears, namely the bottom-set gillnet and collapsible crab trap, used in small-scale fisheries of Bandon Bay, Suratthani Province, Thailand. Both gears mainly target the blue swimming crab (BSC) *Portunus pelagicus*, and together contribute about half of Thailand's annual BSC catch of around 2.5 thousand tonnes. Field sampling was conducted from January to November of 2018. Specimens from bottom-set gillnets and collapsible crab traps comprised 111 and 118 taxa, respectively. Of these, 26 and 27 crab species and 41 and 46 fish species were collected by gillnets and traps, respectively. The index of relative importance of BSC was higher in gillnets (48.8 ± 16.6%) than in traps (25.0 ± 15.5%), where another swimming crab (*Charybdis affinis*) was more common. Cluster analysis revealed that catch compositions were seasonal and differed between the two monsoonal seasons, *i.e.*, northeast monsoon (October to February) and southwest monsoon (May to September), and the transition period (March and April). Potential impact from both fishing gears on various stocks was assessed by standard productivity and susceptibility analysis (PSA). Vulnerability scores of the BSC stock as the main target species suggested it was at moderate risk, as assessed by PSA. The impacts of both gears to stocks of the other species in Bandon Bay showed either low or moderate risk. Ten fish stocks, including two stingrays, six species of sole and two other bony fishes, were near the threshold of high risk from gillnet fishing.

# INTRODUCTION

The Gulf of Thailand (GoT) is one of the world's most productive large marine ecosystems, and it mostly lies within the Thai territory. The total catch from the GoT was around 1.03 million tonnes in 2018, which represented 73% of the country's marine harvest and 42% of the total fisheries and aquaculture production for the year (*Fisheries Development Policy and Planning Division , 2020*). Although the primary fishing targets of marine capture are pelagic and demersal finfishes, three other aquatic animals support valuable fisheries: Indian

squid *Uroteuthis duvauceli*, banana prawn *Penaeus merguiensis* and blue swimming crab (BSC) *Portunus pelagicus* (*Kulanujaree et al., 2020*). Marine fisheries can be characterized as commercial and small-scale fisheries (SSF), of which the latter contributes about 15% of the total marine harvest in Thailand annually (*Derrick et al., 2017*). *Lymer et al. (2008)* mentioned that while the commercial fisheries target multiple species with all gear types, SSF in Thailand, though inevitably capturing a mix of species, are more focused on their target species. This specialization is reflected by the names of the gear; for example, mackerel gillnet, squid falling net and shrimp trammel net. Among the gears used in SSF, two types target crabs (particularly BSC), which are bottom-set gillnets and collapsible crab traps. These two fishing gears, hereafter "gillnets" and "traps", are also used for BSC fisheries elsewhere in the south of Thailand and in other countries of Southeast Asia (*Prince et al., 2020*). In Thailand, the material used for both gears is 2.5 inch (6.4 cm) stretched mesh. Gillnets contain several layers of this mesh, each layer with length of around 180 m and height of 1.25 m. Trap frames are made from aluminum wire with dimensions of 35 × 55 × 17 cm.

Bandon Bay (9°20′00″N, 99°25′00″E; Fig. 1) is in the south of Thailand and home to more than 130 fish species and more than 210 species of other aquatic animals (*Sawusdee, 2010*). The bay area is 477 km$^2$, with 120 km of coastline and mean depth of 2.9 m. Weather patterns are influenced by the northeast and southwest monsoons, which are present almost year-round. Its waters are very productive, owing in part to nutrient inputs from the Tapee River and 18 other river channels (*Jarernpornnipat et al., 2003*; *Sawusdee, 2010*). A 2020 fisheries census in Bandon Bay reported 12,120 fishers, of which 65% were small-scale fishers, operating vessels smaller than 10 gross-tonnes and fishing within 3 nautical miles from shore. The total estimated catch from this bay in 2019 was 31,291 tonnes from almost 30 fishing gear types targeting various groups of aquatic animals (Surat Thani Provincial Fisheries Office, 2020). The substrate of mixed mud, clay and sand, as well as a beach that reaches up to 2 km into the sea, make the bay suitable for numerous crustaceans and other benthic invertebrates, which constitute about 45% of landings from Bandon Bay (*Sawusdee, 2010*; *Plongon & Salaenoi, 2015*). These are reasons the crustaceans are heavily targeted by small-scale fisheries here, making Bandon Bay the primary fishing ground for this aquatic animal group. Of the annual total catch of BSC in Thailand, which averages around 2.5 thousand tonnes, approximately half is from the SSF in Bandon Bay (*Fisheries Development Policy and Planning Division , 2020*). Moreover, this fertile bay is suitable for blood cockle cultivation, and some areas of the bay are dominated by extensive coastal aquaculture of this clam (*Jarernpornnipat et al., 2003*; *Kritsanapuntu & Chaitanawisuti, 2019*).

Fishing gears used in SSF by their nature impact the near-shore ecosystem, where various species of fishes and other aquatic animals reside, either permanently or temporarily. Small-scale fisheries are mostly indiscriminate and may have wide variation in bycatch numbers and rates, and thus, inappropriate operation of these fisheries may negatively impact the abundance, distribution and species composition of vulnerable taxa (*Pinnegar & Engelhard, 2008*; *Shester & Micheli, 2011*). Moreover, the SSF may indirectly impact the ecosystem through habitat degradation, which could cause in decline of megafauna, *e.g.,*
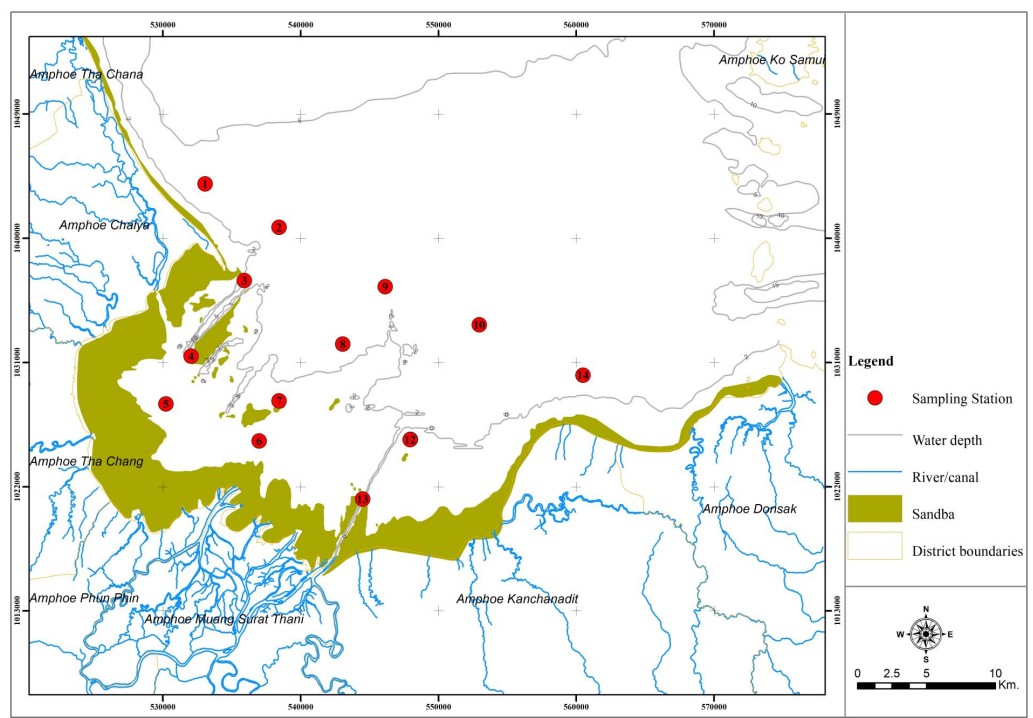

**Figure 1** **Location and map of Bandon Bay, Surratthani, Thailand.** Red dots indicate sampling sites, where fishing gears were deployed.

marine mammals, sea turtles and chondrichthyans (*Temple et al., 2018*). *Shester & Micheli (2011)* reported that ecological impacts by SSF varied according to gear types and habitat characteristics, but that the small size of fishing vessels employed would limit the range of the impacted area. Though SSF are recognized as having low ecological impact on coastal marine resources (*Pauly, 2006*), they still require appropriate management. Importantly, ensuring the sustainable utilization of resources by these fisheries also means supporting the livelihoods and food security of local fishing households (*Smith et al., 2021*). Managing SSF, however, is quite complicated due to the complexity of fishing patterns, which are related to, for example, biogeographic features of the fishing areas, resource availability and fishing gears used (*Coronado et al., 2020*). Also, neither catch nor effort from SSF is included in the official reporting system, making stock assessment difficult and imprecise (*Pita, Villasante & Pascual-Fernández, 2019*; *Song et al., 2020*). Therefore, evaluation of the impact of fishing using a semi-quantitative approach (*i.e.*, Level-2; *Hobday et al., 2011*) is recommended for SSF (*Pita, Villasante & Pascual-Fernández, 2019*). Similar to most of the small-scale fisheries elsewhere, data on the impacts of gillnets and traps used by SSF in Bandon Bay are incomplete, even though the fishery significantly contributes to the country's production of BSC. *Shester & Micheli (2011)* revealed that not only the marine megafauna (mammals, seabirds, and turtles) are threatened by SSF, but also a number of non-target species are impacted by SSF, which have discard rates higher than commercial fisheries. Capacity to withstand fishing intensity varies by species (*Purcell et al., 2018*); thus,

the vulnerability of both target species and non-target species must be known and integrated into fisheries management. This study, therefore, (i) examines the catch composition from gillnets and traps used by SSF in Bandon Bay, and (ii) evaluates the ecological risk of species vulnerable to each type of net. This work also complies with the UN's announcement of 2022 as the Year of Artisanal Fisheries and Aquaculture and the indicator of UN-SDG-14 in securing sustainable small-scale fisheries.

## MATERIALS & METHODS

### Sampling stations and protocol

The Institute of Animals for Scientific Purposes Development approval for this research (U1-04118-2559). Field experiments were approved by Agricultural Research Development Agency (public organization) (project number: PRP6005010660). Fourteen (14) sampling stations were established throughout Bandon Bay, along three longitudinal transects perpendicular to the shoreline and two additional stations at the mouth of the bay. All stations were at least 3 km apart (Fig. 1). Sampling was conducted once a month in every sampling station, from January to November 2018, during a spring tide and using the same sampling protocol. Sampling in December was skipped because of the effects of tropical cyclone "Plabuk". Gillnets and traps used in the field sampling are as explained in the Introduction. On each sampling day at 17:00, three (3) tiers of gill nets and 90 traps were deployed at each sampling station and soaked for 12 h before being recovered. All catches were taken back to the fish landing sites.

### Catch composition analysis

Catches were ice-packed individually and taken back to Walailak University, 160 km from Bandon Bay. At the laboratory, the catches from each station and gear were identified taxonomically (in some cases only to genus or family level), and then weighed and counted. Taxonomy was based on *Nelson, Grande & Wilson (2016)* and FishBase (http://www.fishbase.org; *Froese & Pauly, 2021*) for fishes and *Carpenter & Niem (2001)* and SeaLifeBase (http://www.sealifebase.org; *Palomares & Pauly, 2021*) for other aquatic animals.

The index of relative importance (%IRI) (*Caddy & Sharp, 1986*) was used to express the contribution of individual species in the catches in each month, and calculated as

$$\%IRI = 100 \times \left[ (\%W_i + \%N_i) \times \%F_i \right] \Big/ \left[ \sum \left( (\%W_j + \%N_j) \times \%F_j \right) \right]$$

where %W and %N are the percentages by weight and number of each species $i$ in the total catch, %F is the percentage of occurrence of each species in the total sample, and the denominator is the total of all species $j$. Mann–Whitney $U$ test was applied to examine whether the %IRI of BSC was significantly different between gears. Similarity of the 20 first species of highest %IRI of each gear among sampling months was graphically expressed by dendrogram cluster analysis, using Bray–Curtis dissimilarity matrix and average method. Analysis of similarity (ANOSIM) was used to test similarity among clusters. The data analysis was conducted by using R (*R Core Team, 2021*).

## Risk assessment

Productivity Susceptibility Analysis (PSA; *Hobday et al., 2011*), which is a practical semi-quantitative vulnerability assessment tool (*Hordyk & Carruthers, 2018*; *Lin et al., 2020*; *Faruque & Matsuda, 2021*) was used for assessing the risk of individual stocks from the BSC fisheries in Bandon Bay. The PSA consists of the attributes of two characters: (i) productivity, for determining the rate at which the species can recover from fishing and (ii) susceptibility, for determining the impact to the species caused by fishing. There were seven productivity attributes and four susceptibility attributes used in this study (Table 1). For each species, the data and information for each productivity attribute was from desk study of relevant reports from the GoT and from FishBase (*Froese & Pauly, 2021*) and SeaLifeBase (*Palomares & Pauly, 2021*). In cases where age and size at maturity were not available but growth parameters were, the models were calculated using estimates of the attributes, as proposed by *Froese & Binohlan (2000)*. Meanwhile, the information for each susceptibility attribute was from the observations and results of field sampling for catch composition, desk study, and meetings with experts (*i.e.*, fishery scientists and fishers). The obtained data and information was converted to a rank score (Table 1), where 1 is high productivity or low susceptibility, 2 is medium productivity or susceptibility, and 3 is low productivity or high susceptibility (*Hordyk & Carruthers, 2018*). It is worth noting that the rank scores for productivity attributes are adjusted to be suitable for tropical aquatic taxa (*FAO, 2014*). A focus group discussion among the researchers, fisheries scientists and fishers was conducted to discuss the rank scores of the catches, and in particular, maximum and maturity sizes, selectivity of gear types, as well as abundance and occurrence of individual species in the studied area. This activity was included in the study so that fisheries scientists and fishers could provide expert judgment, fishery-specific experienceand ecological knowledge relevant to each attribute (*Hobday et al., 2011*). The total vulnerability ($V$) or risk score was then calculated by

$$V = \sqrt{P^2 + S^2}$$

where $P$ is the overall productivity score (*i.e.*, arithmetic mean of the productivity attributes) and S is the overall susceptibility score (*i.e.*, geometric mean of the susceptibility attributes). The *V score* ranges between 1.41 and 4.24; values lower than 2.64 and above 3.18 are considered low and high vulnerability, respectively, while values in between indicate medium vulnerability (*Hobday et al., 2011*; *Hordyk & Carruthers, 2018*).

A data quality score (Table 2) was also estimated for each species for interpretation of the vulnerability scores (*Patrick et al., 2010*; *Ormseth & Spencer, 2011*; *Faruque & Matsuda, 2021*). The mean quality score of P and S was interpreted as high (<2), medium (≥2 and <3), or low (≥3). Difference in *V score* s between the two fishing gears for each species (or higher taxon) was tested by Mann–Whitney $U$ test. All statistical tests were conducted by using R (*R Core Team, 2021*).

## RESULTS

In total, the sampled animals comprised 7,880 individuals with a weight of 246,747 g. Catch compositions by percentages in numbers and weight are shown in Fig. 2, meanwhile
**Table 1  List of attributes used for productivity analysis (a) and susceptibility analysis (b) of the BSC fisheries in Bandon Bay.**

**(a) Productivity**

| Productivity attributes | Productivity/Risk | | |
| --- | --- | --- | --- |
| | Low productivity/ High risk (Score = 3) | Medium productivity/ Medium risk (Score = 2) | High productivity/ Low risk (Score = 1) |
| Average age at maturity (years) | >4 | 2 to 4 | <2 |
| Average maximum age (years) | >30 | 10 to 30 | <10 |
| Fecundity (eggs/spawning) | <1,000 | 1,000 to 10,000 | >10,000 |
| Average maximum size (cm) | >150 | 60 to 150 | <60 |
| Average size at maturity (cm) | >150 | 30 to 150 | <30 |
| Reproductive strategy | Live bearer, mouth brooder or significant parental investment | Demersal spawner or "berried" | Broadcast spawner |
| Mean trophic level | >3.25 | 2.5 –3.25 | <3.25 |

**Susceptibility**

| Susceptibility attributes | Susceptibility/Risk | | |
| --- | --- | --- | --- |
| | High risk (Score = 3) | Medium risk (Score = 2) | Low risk (Score = 1) |
| Availability I: Overlap of adult species range with fishery | >50% of stock occurs in the area fished | 25% and 50% of stock occurs in the area fished | <25% of stock occurs in the area fished |
| Availability II: Distribution | Only in the country/ fishery | Limited range in the region | Throughout the region/global |
| Encounterability I: Habitat | Habitat preference of species make it highly likely to encounter gears | Habitat preference of species make it moderately likely to encounter gears | Depth or distribution of species make it unlikely to encounter gears |
| Encounterability II: Depth range | High overlap with fishing gears | Medium overlap with fishing gears | Low overlap with fishing gears |
| Selectivity | Species >2 times mesh size | Species 1 or 2 >mesh size | Species <mesh size or too large to be selected |
| Post capture mortality | Probability of survival <33% | Between 33% and 67% probability of survival | Probability of survival >67% |

**Table 2  Rank scores for data quality used for the productivity-susceptibility analysis of the blue swimming crab fisheries in Bandon Bay, Suratthani, Thailand.**

| Score | Data quality | Description |
| --- | --- | --- |
| 1 | Best data | Information is based on collected data for the stock and area of interest that is established and substantial |
| 2 | Adequate data | Information is based on limited coverage and corroboration, or for some other reason is deemed not as reliable as tier-1 data |
| 3 | Limited data | Estimates with high variation and limited confidence, and may be based on studies of similar taxa or life history strategies |
| 4 | Very limited data | Information based on expert opinion or general literature reviews from a wide range of species, or from outside of region, or data derived by equation using the correlated life history parameters |
| 5 | No data | No information available |

percentages of individual species are presented in Table 3. There were 111 and 118 species of fish and other aquatic animals caught by gillnets and traps, respectively (Table 3). No endangered, threatened or protected (ETP) species were included in the catch composition throughout the study. Similar groups of marine invertebrates were caught in both fishing gears, albeit with some difference at genus or species levels. There were 26 and 27 species of crab (Families Diogenidae, Dorippidae, Leucosiidae, Matutidae, Epialtidae, Galenidae, Parthenopidae, Portunidae, Menippidae Galenidae Macrophthalmidae and Varunidae) caught by gillnets and traps, respectively. Other marketable aquatic animals caught by both gears included gastropods, bivalves, cephalopods, mantis shrimps and sea cucumbers. Over 40 fish species, both teleost and elasmobranch, a were collected throughout the study (41 by gillnets and 46 by traps). Some species groups were retained in a particular gear, for example, sting rays were caught only by gillnets, while gobies were found only in traps. The five most commonly caught species by number in gillnets were gastropod *Murex* sp. (26.6%), followed by BSC (22.2%), crab *Dorippe quadridens* (7.0%), sea urchin *Temnopleurus toreumaticus* (6.5%) and crab *Macrophthalmus* sp. (4.9%). Meanwhile, three out of the five most common species, by number, in traps were crabs, *Charybdis affinis* (37.2%), BSC (11.1%), and *D. quadridens* (4.1%), followed by *T. toreumaticus* (1.6%) and hermit crab *Clibanarius infraspinatus* (1.6%). In terms of weight, BSC was ranked first for both gears, and contributed over 50% in gillnets and about 27% in traps. Another species of swimming crab, *C. affinis*, was also common in traps; if its weight was added with BSC, their percentage would be over 50% of the catch. Notably, the two species in each gear with the highest overall mean %IRI had values over 15%; meanwhile, the remaining taxa were less than 5% (Table 3). Overall means ($\pm$ SD) of %IRI for BSC in gillnets (48.8 $\pm$ 16.6%) and traps (25.0 $\pm$ 15.5%) were statistically different (Mann–Whitney $U$ test, $P = 0.005$; Fig. 3). Dendrogram clusters for each month showed that BSC was by far the dominant species in terms of %IRI in gillnets, followed by *Murex* sp. (Fig. 4A). However, in traps, *C. affinis* was ranked first in %IRI, followed by BSC (Fig. 4B). Catch compositions differed seasonally and were separated into three distinct clusters for each gear (ANOSIM, $P < 0.02$). Higher numbers of species were found in the catch during summer (March to April) in both gears. For gillnets, BSC dominated the catches during the northeast monsoon (October to February), while *Murex* sp. showed higher %IRI during the southwest monsoon (May to September). Meanwhile, highest %IRI for BSC in traps was observed during the southwest monsoon.

Data quality scores for the productivity attributes ranged between 1.0 and 4.0, with an average of 1.8 $\pm$ 1.4, implying relatively high quality of information used to interpret the vulnerability of stocks of fish and other aquatic animals to the Bandon Bay BSC fisheries. Vulnerability ($V$) scores of individual species for both gears are presented in Table 3. The overall $V$ score ranged from 1.81 to 3.16 (2.78 $\pm$ 0.28) for gillnets and from 1.70 to 2.93 (2.29 $\pm$ 0.33) for traps. Results indicated that the BSC was at moderate risk ($V = 2.86$) from both gears, for which the $P$ and $S$ scores were 1.14 and 2.62, respectively. Eighty (80) species were at moderate risk from the gillnet fishery; meanwhile, the majority of species that are catchable by trap (96 out of 118 stocks) faced low risk from the trap fishery, *i.e.*, $V$ score lower than 2.64. Although no species were rated as high risk from BSC gillnets or

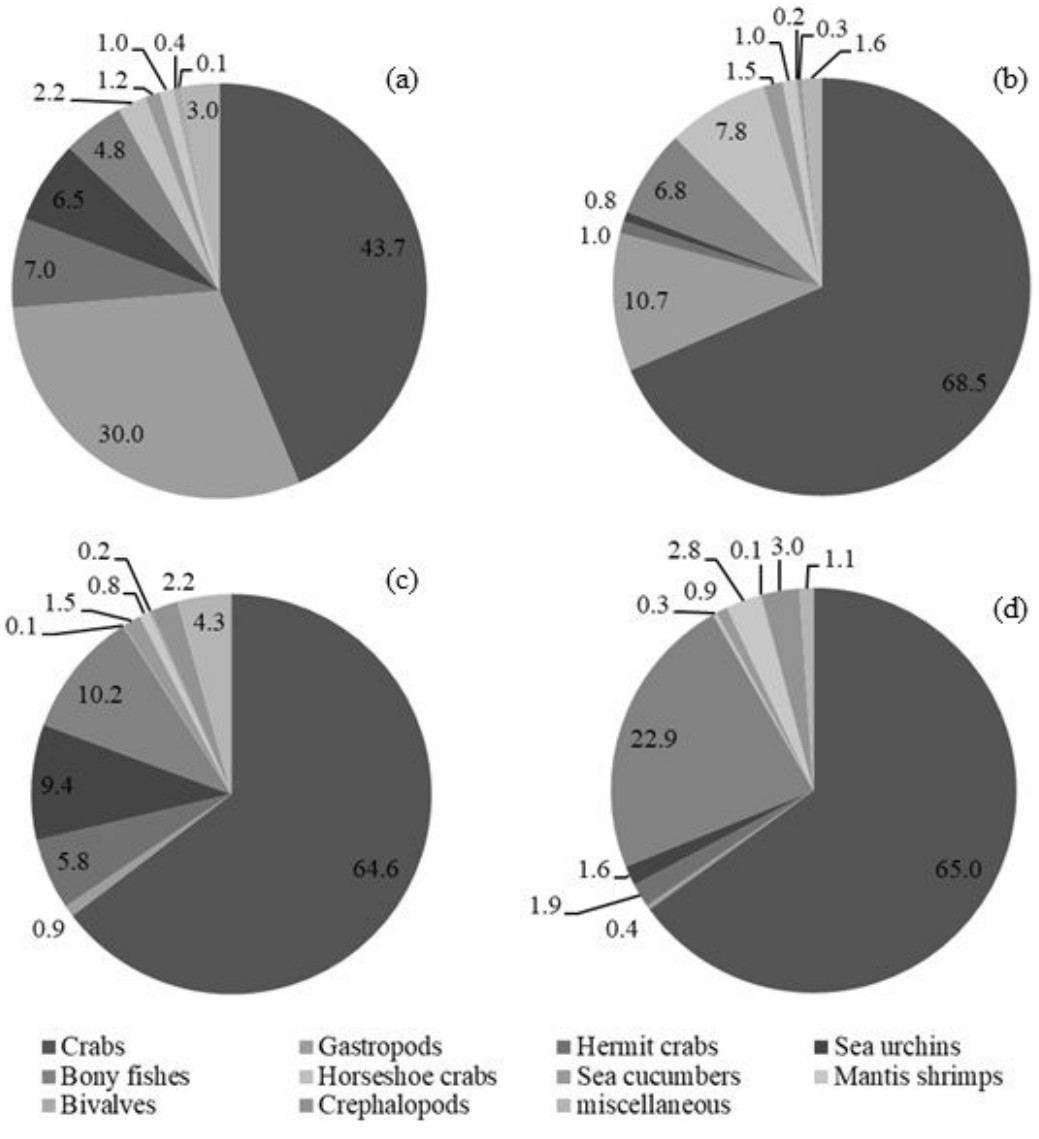

**Figure 2** Catch composition by percentages of (A) number and (B) weight in bottom-set gillnets and by percentages of (C) number and (D) weight in collapsible crab traps in Bandon Bay, Surratthani, Thailand.

traps in Bandon Bay, there were 10 fish species with high $V\,score$s (*i.e.*, near the threshold of 3.18) in the gillnet fishery. These fishes included two elasmobranchs (*Himantura imbricate* and *Maculabatis gerrardi*), two bony fishes (*Muraenesox cinereus* and *Hexanematichthys sagor*) and a group of sole species (Family Soleidae and Cynoglossidae). A graphical PSA of selected individual stocks and stock-groups, which are marketed species, from gillnet and trap fisheries in Bandon Bay is presented in Fig. 5. Results (Fig. 6) revealed that there were non-significant differences between gears in levels of risk to bivalves (Mann–Whitney $U$ test, $P = 0.55$), cephalopods (Mann–Whitney $U$ test, $P = 0.47$) and mantis shrimp (Mann–Whitney $U$ test, $P = 0.05$). However, significant differences were found for

gastropods (Mann–Whitney $U$ test, $P$-values <0.001), prawns (Mann–Whitney $U$ test, $P = 0.04$), crabs (Mann–Whitney $U$ test, $P < 0.001$), sea cucumbers (Mann–Whitney $U$ test, $P = 0.03$), and bony fishes ((Mann–Whitney $U$ test, $P < 0.01$)), for which more risk was found from the gillnet fishery. By averaging the $V$ $score$ s of both fishing gears (Table 3), results revealed that 57 species were at medium risk, as their $V$ scores were between 2.64 and 3.18, from the SSF of Bandon Bay.

## DISCUSSION

Results of this study confirm the indiscriminate nature in terms of catch composition of the small-scale gillnet and trap fisheries of the productive Bandon Bay in the Gulf of Thailand. Risks by SSF are overlooked in assessments, which generally focus on commercial fisheries. This is unsurprising, as the uneven history of fisheries science was not conceived for multi-species SSF (*Smith et al., 2021*). Similar to most of the small-scale coastal fisheries elsewhere in the tropics, catches from the SSF of Bandon Bay are multi-species due to the productivity of the area and captured from both fisheries in Bandon Bay is considerably lower than the 170 species diversity of aquatic animals inhabiting this fishing ground. The roughly 100 species collected from the gillnet SSF in Pattani Bay, lower Gulf of Thailand (*Fazrul et al., 2015*). Meanwhile, there were 45 and 77 species of fishes and other aquatic animals collected from gillnet and trap SSF (which also target BSC) at Phu Quoc Island, Vietnam (*Ha et al., 2015*); however, no bivalves, starfish, mantis shrimp, horseshoe crabs or sea cucumbers were mentioned in the report. The number of crab species in SSF in Thai waters has ranged between 17 and 27, in which the mud crab *Scylla* spp. and crab *Charybdis* spp. are also market-valued species and can be caught in substantial numbers, comparable to BSC (*Fazrul et al., 2015*; *Kunsook & Dumrongrojwatthana, 2017*; this study). Attempts to reduce the non-targeted catch in these two fishing gears include a proposal to not allow gillnets to be operated in near-shore areas for a fishery in Indonesia (*Supadminingsih, Riyanto & Wahju, 2018*). *Boutson et al. (2009)* reported that a trap with escape vents could potentially reduce the number of non-target species; however, the number of the targeted BSC captured by the trap with escape vents was about three times lower than the conventional one, which would likely not be accepted by fishers.

Crabs, in particular BSC, remained a high proportion of the catch in both gears throughout the study period in Bandon Bay. It was observed during our samplings that most of the BSC caught were larger over 10 cm in outer carapace width (OCW), which is slightly above the size at 50% maturity of about 9.5 cm OCW (*Nillrat et al., 2019*). The peak BSC catch in BSC fisheries in South Sulawesi, Indonesia, was observed from May to September and not during the two rainy seasons, which are from January to April and from November to December (*Wiyono & Ihsan, 2018*). In this study, the %IRI of BSC in traps dropped during the northeast monsoon (November to February); meanwhile, %IRI of BSC in gillnets dropped from April to June. Because Bandon Bay is relatively shallow, water turbulence during the monsoon would make the crabs and other aquatic animals less gregarious and increase habitat rugosity, factors which are both negatively correlated with catchability by traps (*Robichaud, Hunte & Chapman, 2000*). Moreover, the turbulence

Jutagate and Sawusdee (2022), *PeerJ*, DOI 10.7717/peerj.13878
**Table 3  List of taxa captured, their contribution in catches and risks in the small-scale fisheries of the Bandon Bay, Thailand.**

| Family | Scientific name | %N (G) | %W (G) | %N (T) | %W (T) | %IRI (G) | %IRI (T) | P | QP | S (G) | V (G) | S (T) | V (T) |
|---|---|---|---|---|---|---|---|---|---|---|---|---|---|
| Actiniidae | *Anthopleura* sp. | 0.30 | 0.03 | 0.13 | 0.01 | 0.05 | 0.01 | NA | 4.14 | 1.26 | NA | 1.26 | NA |
| Strombidae | *Doxander vittatus* | 0.04 | <0.01 | NA | NA | 0.01 | NA | 1.14 | 3.57 | 2.62 | 2.86 | NA | NA |
| Bursidae | *Bufonaria crumena* | 0.22 | 0.12 | NA | NA | 0.10 | NA | 1.14 | 3.57 | 2.62 | 2.86 | NA | NA |
| Naticidae | *Natica vitellus* | NA | NA | <0.01 | <0.01 | NA | <0.01 | 1.14 | 2.57 | NA | NA | 1.70 | 2.05 |
| Muricidae | *Lataxiena blosvillei* | NA | NA | <0.01 | <0.01 | NA | <0.01 | 1.14 | 3.57 | NA | NA | 1.70 | 2.05 |
| Muricidae | *Murex trapa* | 0.04 | 0.01 | NA | NA | 0.02 | NA | 1.14 | 2.57 | 2.62 | 2.86 | NA | NA |
| Muricidae | *Murex* sp.1 | 26.60 | 4.23 | 0.07 | 0.02 | 17.69 | <0.01 | 1.14 | 2.57 | 2.62 | 2.86 | 1.70 | 2.05 |
| Muricidae | *Murex* sp.2 | 1.09 | 0.38 | 0.02 | 0.01 | 0.40 | <0.01 | 1.14 | 2.57 | 2.62 | 2.86 | 1.91 | 2.23 |
| Muricidae | *Indothais* sp. | 0.22 | 0.04 | 0.48 | 0.05 | 0.07 | 0.01 | 1.14 | 2.57 | 2.45 | 2.70 | 1.70 | 2.05 |
| Nassariidae | *Rapana rapiformis* | 0.04 | 0.10 | NA | NA | 0.01 | NA | 1.14 | 2.57 | 2.62 | 2.86 | NA | NA |
| Nassariidae | *Nassaria pusilla* | 0.09 | <0.01 | 0.23 | 0.11 | 0.01 | 0.01 | 1.14 | 3.71 | 2.62 | 2.86 | 1.70 | 2.05 |
| Nassariidae | *Nassarius siquijorensis* | NA | NA | 0.04 | 0.02 | NA | <0.01 | 1.14 | 3.71 | NA | NA | 1.70 | 2.05 |
| Melongenidae | *Hemifusus* sp. | 0.43 | 0.35 | 0.04 | 0.03 | 0.19 | <0.01 | 1.14 | 3.14 | 2.62 | 2.86 | 2.04 | 2.34 |
| Melongenidae | *Pugilina Schumacher* | 0.96 | 2.64 | 0.02 | 0.03 | 1.46 | <0.01 | 1.14 | 3.14 | 2.62 | 2.86 | 1.70 | 2.05 |
| Fasciolariidae | *Pleuroploca* sp. | NA | NA | <0.01 | <0.01 | NA | <0.01 | 1.14 | 3.71 | NA | NA | 1.91 | 2.23 |
| Volutidae | *Cymbiola nobilis* | 0.04 | 0.89 | 0.02 | 0.11 | 0.16 | <0.01 | 1.14 | 1.86 | 2.45 | 2.7 | 1.70 | 2.05 |
| Volutidae | *Melo melo* | 0.17 | 1.91 | NA | NA | 0.40 | NA | 1.14 | 1.86 | 2.62 | 2.86 | NA | NA |
| Arcidae | *Anadara inaequivalvis* | 0.09 | 0.13 | 0.11 | 0.08 | 0.04 | <0.01 | 1.00 | 2.71 | 1.82 | 2.07 | 1.70 | 1.97 |
| Arcidae | *Tegillarca nodifera* | 0.30 | 0.03 | 0.09 | 0.02 | 0.07 | <0.01 | 1.00 | 2.71 | 1.82 | 2.07 | 1.70 | 1.97 |
| Pectinidae | *Chlamys* sp. | NA | NA | 0.02 | <0.01 | NA | <0.01 | 1.00 | 3.14 | NA | NA | 1.70 | 1.97 |
| Pectinidae | *Mimachlamys* sp. | 0.04 | 0.01 | 0.02 | <0.01 | 0.01 | <0.01 | 1.00 | 2.57 | 1.51 | 1.81 | 1.70 | 1.97 |
| Sepiidae | *Sepia* sp.1 | 0.09 | 0.26 | 0.66 | 1.17 | 0.05 | 0.15 | 1.57 | 1.71 | 2.04 | 2.57 | 2.00 | 2.54 |
| Sepiidae | *Sepia* sp.2 | NA | NA | 0.36 | 0.70 | NA | 0.08 | 1.57 | 1.71 | NA | NA | 2.00 | 2.54 |
| Sepiidae | *Sepiella inermis* | NA | NA | 1.17 | 1.08 | NA | 0.56 | 1.57 | 1.71 | NA | NA | 1.78 | 2.37 |
| Octopodidae | *Octopus* sp. | 0.04 | 0.02 | 0.04 | 0.10 | 0.01 | <0.01 | 1.57 | 1.57 | 1.94 | 2.5 | 1.78 | 2.37 |
| Limulidae | *Carcinoscorpius rotundicauda* | 0.35 | 0.54 | <0.01 | <0.01 | 0.24 | <0.01 | 1.71 | 2.14 | 2.45 | 2.99 | NA | NA |
| Limulidae | *Tachypleus gigas* | 1.87 | 7.30 | 0.05 | 0.32 | 4.78 | 0.02 | 1.71 | 2.14 | 2.62 | 3.13 | 1.70 | 2.41 |
| Squillidae | *Harpiosquilla harpax* | 0.26 | 0.48 | 0.47 | 1.59 | 0.18 | 0.24 | 1.29 | 1.86 | 2.45 | 2.77 | 1.91 | 2.30 |
| Squillidae | *Harpiosquilla raphidea* | 0.04 | 0.09 | 0.13 | 0.47 | 0.03 | 0.02 | 1.29 | 1.86 | 2.29 | 2.63 | 2.14 | 2.50 |
| Squillidae | *Oratosquillina interrupta* | 0.35 | 0.18 | 0.09 | 0.64 | 0.12 | 0.01 | 1.29 | 1.86 | 2.29 | 2.63 | 2.29 | 2.63 |
| Squillidae | *Oratosquilla nepa* | 0.39 | 0.24 | 0.05 | 0.07 | 0.31 | <0.01 | 1.29 | 1.86 | 2.45 | 2.77 | 2.18 | 2.53 |
| Squillidae | *Oratosquilla woodmasoni* | NA | NA | 0.04 | 0.01 | NA | <0.01 | 1.29 | 1.86 | NA | NA | 2.29 | 2.63 |
| Scyllaridae | *Thenus indicus* | 0.13 | 0.31 | NA | NA | 0.17 | NA | 1.29 | 3.43 | 2.62 | 2.92 | NA | NA |
| Penaeidae | *Metapenaeus* sp. | NA | NA | 0.04 | <0.01 | NA | <0.01 | 1.14 | 1.14 | NA | NA | 2.04 | 2.34 |
| Penaeidae | *Penaeus semisulcatus* | <0.01 | <0.01 | 0.04 | 0.01 | <0.01 | <0.01 | 1.14 | 1.14 | 2.80 | 3.03 | 1.91 | 2.23 |

Jutagate and Sawusdee (2022), *PeerJ*, DOI 10.7717/peerj.13878

**Table 3** (*continued*)

| Family | Scientific name | % N (G) | %W (G) | %N (T) | %W (T) | %IRI (G) | %IRI (T) | P | QP | S (G) | V (G) | S (T) | V (T) |
|---|---|---|---|---|---|---|---|---|---|---|---|---|---|
| Penaeidae | *Penaeus silasi* | NA | NA | 0.07 | 0.04 | NA | <0.01 | 1.14 | 1.14 | NA | NA | 2.04 | 2.34 |
| Palaemonidae | *Macrobrachium rosenbergii* | NA | NA | 0.02 | 0.06 | NA | <0.01 | 1.29 | 1.14 | NA | NA | 1.41 | 1.91 |
| Diogenidae | *Diogenes* sp.1 | 1.13 | 0.06 | 1.04 | 0.23 | 0.36 | 0.19 | 1.29 | 2.71 | 2.62 | 2.92 | 2.18 | 2.53 |
| Diogenidae | *Diogenes* sp.2 | 4.65 | 0.37 | 0.30 | 0.02 | 2.33 | 0.03 | 1.29 | 2.71 | 2.62 | 2.92 | 2.04 | 2.41 |
| Diogenidae | *Clibanarius infraspinatus* | 1.17 | 0.53 | 4.30 | 1.57 | 0.55 | 0.96 | 1.29 | 2.71 | 2.45 | 2.77 | 2.45 | 2.77 |
| Diogenidae | *Dardanus lagopodes* | NA | NA | 0.13 | 0.07 | NA | <0.01 | 1.29 | 2.71 | NA | NA | 2.29 | 2.63 |
| Dorippidae | *Dorippe quadridens* | 7.04 | 1.78 | 10.77 | 4.08 | 4.52 | 4.92 | 1.14 | 2.00 | 2.62 | 2.86 | 2.45 | 2.70 |
| Dorippidae | *Neodorippe callida* | NA | NA | 0.02 | <0.01 | NA | <0.01 | 1.14 | 2.16 | NA | NA | 2.29 | 2.56 |
| Leucosiidae | *Seulocia vittata* | 1.74 | 0.2 | 0.39 | 0.02 | 0.66 | NA | NA | 4.00 | 2.62 | NA | 2.45 | NA |
| Matutidae | *Matuta planipes* | 0.04 | 0.05 | 0.22 | 0.20 | 0.02 | 0.01 | 1.29 | 2.29 | 2.62 | 2.92 | 2.04 | 2.41 |
| Matutidae | *Matuta victor* | <0.01 | <0.01 | 0.22 | 0.13 | <0.01 | 0.02 | 1.29 | 2.29 | 2.62 | 2.92 | 2.04 | 2.41 |
| Epialtidae | *Doclea armata* | 0.30 | 0.10 | 0.30 | 0.03 | 0.09 | 0.04 | 1.14 | 2.86 | 2.62 | 2.86 | 2.62 | 2.86 |
| Epialtidae | *Doclea canalifera* | 0.65 | 1.33 | 0.34 | 0.29 | 1.28 | 0.04 | 1.14 | 2.86 | 2.62 | 2.86 | 2.62 | 2.86 |
| Epialtidae | *Doclea rissoni* | NA | NA | 0.13 | 0.11 | NA | 0.01 | 1.14 | 2.86 | NA | NA | 2.45 | 2.70 |
| Epialtidae | *Doclea* sp. | 0.26 | 0.16 | 0.95 | 0.72 | 0.08 | 0.16 | 1.14 | 2.86 | 2.62 | 2.86 | 2.18 | 2.46 |
| Galenidae | *Galene bispinosa* | 0.26 | 0.41 | 0.02 | 0.07 | 0.17 | <0.01 | 1.29 | 4.00 | 2.62 | 2.92 | 2.18 | 2.53 |
| Galenidae | *Halimede ochtodes* | 0.26 | 0.17 | NA | NA | 0.15 | NA | 1.29 | 4.00 | 2.62 | 2.92 | NA | NA |
| Parthenopidae | *Rhinolambrus* sp. | 0.70 | 0.26 | NA | NA | 0.21 | NA | NA | 4.14 | 2.62 | NA | NA | NA |
| Portunidae | *Lupocycloporus gracilimanus* | NA | NA | <0.01 | <0.01 | NA | <0.01 | 1.14 | 1.00 | NA | NA | 2.45 | 2.70 |
| Portunidae | *Portunus haanii* | 0.04 | 0.01 | <0.01 | <0.01 | 0.01 | <0.01 | 1.14 | 1.00 | 2.62 | 2.86 | 2.62 | 2.86 |
| Portunidae | *Portunus pelagicus* | 22.21 | 58.65 | 11.08 | 26.84 | 48.85 | 24.98 | 1.14 | 1.00 | 2.62 | 2.86 | 2.62 | 2.86 |
| Portunidae | *Portunus sanguinolentus* | 0.48 | 1.08 | 0.13 | 0.09 | 0.46 | 0.02 | 1.14 | 1.00 | 2.62 | 2.86 | 2.45 | 2.7 |
| Portunidae | *Scylla olivacea* | NA | NA | 0.04 | 0.65 | NA | 0.01 | 1.14 | 1.00 | NA | NA | 2.45 | 2.7 |
| Portunidae | *Xiphonectes hastatoides* | 0.04 | 0.01 | NA | NA | 0.01 | NA | 1.14 | 1.00 | 2.62 | 2.86 | NA | NA |
| Portunidae | *Charybdis affinis* | 3.52 | 1.52 | 37.16 | 24.14 | 1.98 | 56.61 | 1.29 | 1.86 | 2.45 | 2.77 | 2.62 | 2.92 |
| Portunidae | *Charybdis anisodon* | 0.74 | 0.29 | 0.32 | 0.15 | 0.47 | 0.04 | 1.29 | 1.86 | 2.62 | 2.92 | 2.18 | 2.53 |
| Portunidae | *Charybdis feriata* | 0.13 | 0.46 | 0.91 | 3.82 | 0.15 | 0.68 | 1.29 | 1.86 | 2.62 | 2.92 | 2.45 | 2.77 |
| Portunidae | *Charybdis natator* | 0.09 | 0.31 | NA | NA | 0.09 | NA | 1.29 | 1.86 | 2.62 | 2.92 | NA | NA |
| Portunidae | *Charybdis truncata* | NA | NA | 0.02 | <0.01 | NA | <0.01 | 1.29 | 1.86 | NA | NA | 2.62 | 2.92 |
| Portunidae | *Thalamita crenata* | NA | NA | <0.01 | <0.01 | NA | <0.01 | 1.14 | 1.86 | NA | NA | 2.29 | 2.56 |
| Portunidae | *Thalamita spinimana* | 0.04 | 0.05 | 0.70 | 0.84 | 0.01 | 0.10 | 1.14 | 1.00 | 2.62 | 2.86 | 2.29 | 2.56 |
| Portunidae | *Thalamita sima* | NA | NA | 0.13 | 0.13 | NA | 0.01 | 1.14 | 1.86 | NA | NA | 2.29 | 2.56 |
| Portunidae | *Podophthalmus vigil* | <0.01 | <0.01 | NA | NA | <0.01 | NA | 1.14 | 1.00 | 2.62 | 2.86 | NA | NA |
| Menippidae | *Myomenippe hardwickii* | 0.13 | 0.08 | 0.65 | 2.48 | 0.02 | 0.24 | 1.14 | 4.14 | 2.62 | NA | 2.45 | 2.7 |
| Galenidae | *Halimede ochtodes* | NA | NA | 0.09 | 0.13 | NA | 0.01 | 1.29 | 4.14 | NA | NA | 2.29 | 2.63 |
| Macrophthalmidae | *Macrophthalmus* sp. | 4.91 | 1.25 | NA | NA | 1.88 | NA | 1.50 | 3.57 | 2.62 | 3.02 | NA | NA |
| Varunidae | *Varuna yui* | NA | NA | <0.01 | 0.08 | NA | <0.01 | NA | 3.57 | NA | NA | 2.18 | NA |

**Table 3** (*continued*)

| Family | Scientific name | % N (G) | %W (G) | %N (T) | %W (T) | %IRI (G) | %IRI (T) | P | QP | S (G) | V (G) | S (T) | V (T) |
|---|---|---|---|---|---|---|---|---|---|---|---|---|---|
| Ophiotrichidae | *Ophiocnemis marmorata* | <0.01 | <0.01 | NA | NA | <0.01 | NA | NA | 4.00 | 2.80 | NA | NA | NA |
| Ophiotrichidae | *Ophiocnemis* sp. | NA | NA | 0.02 | <0.01 | NA | <0.01 | NA | 3.86 | NA | NA | 2.45 | NA |
| Ophiotrichidae | *Luidia* sp. | 0.04 | 0.02 | 0.63 | 0.34 | 0.02 | 0.14 | NA | 3.86 | 2.45 | NA | 2.45 | NA |
| Astropectinidea | Astropecten sp. 1 | <0.01 | <0.01 | 0.11 | 0.01 | <0.01 | <0.01 | 1.14 | 3.86 | 2.45 | NA | 2.29 | 2.56 |
| Astropectinidea | Astropecten sp. 2 | 1.91 | 0.24 | 2.96 | 0.51 | 0.92 | 0.88 | 1.14 | 3.86 | 2.18 | NA | 2.29 | 2.56 |
| Holothuriidae | *Acaudina* sp.1 | 0.52 | 0.22 | 0.88 | 0.46 | 0.22 | 0.08 | 1.14 | 2.86 | 2.62 | 2.86 | 2.18 | 2.46 |
| Holothuriidae | *Acaudina* sp.2 | 0.13 | 0.18 | 0.22 | 0.06 | 0.09 | 0.01 | 1.14 | 2.86 | 2.62 | 2.86 | 1.70 | 2.05 |
| Phyllophoridae | *Phyllophorella kohkutiensis* | 0.43 | 1.09 | 0.36 | 0.39 | 0.59 | 0.06 | 1.14 | 2.86 | 2.62 | 2.86 | 1.41 | 1.81 |
| Caudinidae | *Holothuria* spp. | 0.09 | 0.01 | 0.04 | <0.01 | 0.01 | <0.01 | 1.14 | 2.86 | 2.45 | 2.70 | 1.70 | 2.05 |
| Pennatulidae | *Pteroeides* sp. | 0.48 | 0.4 | 0.16 | 0.12 | 0.24 | 0.01 | 1.00 | 4.43 | 2.14 | 2.36 | 1.26 | NA |
| Temnopleuridae | *Temnopleurus toreumaticus* | 6.48 | 0.76 | 9.37 | 1.59 | 2.86 | 1.62 | 1.00 | 3.71 | 2.62 | NA | 2.62 | 2.80 |
| Schizasteridae | *Schizaster lacunosus* | 0.04 | 0.02 | NA | NA | 0.01 | NA | 1.50 | 4.33 | 2.29 | 2.74 | NA | NA |
| Clypeasteridae | *Arachnoides placenta* | NA | NA | 0.18 | 0.01 | NA | 0.03 | 1.00 | 3.57 | NA | NA | 1.82 | 2.08 |
| Dasyatidae | *Himantura imbricata* | 0.17 | 0.63 | NA | NA | 0.23 | NA | 2.00 | 1.86 | 2.45 | 3.16 | NA | NA |
| Dasyatidae | *Maculabatis gerrardi* | 0.09 | 0.23 | NA | NA | 0.05 | NA | 2.00 | 2.43 | 2.45 | 3.16 | NA | NA |
| Muraenesocidae | *Muraenesox cinereus* | 0.04 | 0.44 | <0.01 | <0.01 | 0.29 | <0.01 | 2.00 | 1.86 | 2.45 | 3.16 | 1.41 | 2.57 |
| Clupeidae | *Sardinella gibbosa* | NA | NA | 0.13 | 0.04 | NA | 0.01 | 1.14 | 1.71 | NA | NA | 1.41 | 1.81 |
| Engraulidae | *Thryssa kammalensis* | NA | NA | 0.04 | <0.01 | NA | <0.01 | 1.43 | 1.86 | NA | NA | 1.26 | 1.90 |
| Ariidae | *Hexanematichthys sagor* | <0.01 | 0.01 | NA | NA | <0.01 | NA | 2.00 | 1.86 | 2.45 | 3.16 | NA | NA |
| Batrachoididae | *Batrachomoeus trispinosus* | NA | NA | 0.16 | 0.66 | NA | 0.02 | 1.86 | 1.71 | NA | NA | 1.78 | 2.57 |
| Syngnathidae | *Hippocampus* sp. | 0.04 | <0.01 | NA | NA | <0.01 | NA | 1.86 | 2.71 | 1.70 | 2.52 | NA | NA |
| Tetrarogidae | *Vespicula trachinoides* | NA | NA | 0.16 | 0.02 | NA | 0.04 | 1.57 | 2.00 | NA | NA | 1.26 | 2.01 |
| Platycephalidae | *Platycephalus indicus* | 0.09 | 0.24 | 0.07 | 0.10 | 0.19 | <0.01 | 1.57 | 1.86 | 2.62 | 3.06 | 1.59 | 2.24 |
| Platycephalidae | *Platycephalus* sp. | 0.61 | 1.43 | NA | NA | 0.92 | NA | 1.57 | 2.29 | 2.62 | 3.06 | NA | NA |
| Ambassidae | *Ambassis* sp. | NA | NA | 0.23 | 0.01 | NA | 0.01 | 1.29 | 1.86 | NA | NA | 1.26 | 1.80 |
| Serranidae | *Epinephelus coioides* | NA | NA | <0.01 | <0.01 | NA | <0.01 | 2.00 | 1.71 | NA | NA | 1.26 | 2.36 |
| Serranidae | *Epinephelus sexfasciatus* | NA | NA | 0.04 | 0.06 | NA | <0.01 | 1.43 | 1.86 | NA | NA | 1.26 | 1.90 |
| Teraponidae | *Terapon jarbua* | NA | NA | 0.32 | 0.13 | NA | 0.05 | 1.57 | 1.86 | NA | NA | 1.26 | 2.01 |
| Teraponidae | *Terapon puta* | 0.04 | 0.03 | 0.25 | 0.03 | 0.01 | 0.05 | 1.14 | 1.86 | 2.62 | 2.86 | 1.59 | 1.96 |
| Teraponidae | *Terapon theraps* | NA | NA | 0.11 | 0.01 | NA | 0.01 | 1.29 | 2.00 | NA | NA | 1.59 | 2.04 |
| Priacanthidae | *Priacanthus tayenus* | 0.09 | 0.07 | NA | NA | 0.03 | NA | 1.29 | 1.86 | 1.94 | 2.33 | NA | NA |

**Table 3** (*continued*)

| Family | Scientific name | % N (G) | %W (G) | %N (T) | %W (T) | %IRI (G) | %IRI (T) | P | QP | S (G) | V (G) | S (T) | V (T) |
|---|---|---|---|---|---|---|---|---|---|---|---|---|---|
| Apogonidae | *Ostorhinchus fasciatus* | NA | NA | <0.01 | <0.01 | NA | <0.01 | 1.71 | 2.14 | NA | NA | 1.26 | 2.13 |
| Sillaginidae | *Sillago sihama* | 0.09 | <0.01 | NA | NA | <0.01 | NA | 1.29 | 1.86 | 2.18 | 2.53 | NA | NA |
| Carangidae | *Alepes djedaba* | NA | NA | 0.25 | 0.05 | NA | 0.05 | 1.43 | 2.00 | NA | NA | 1.41 | 2.01 |
| Carangidae | *Carangoides praeustus* | NA | NA | <0.01 | <0.01 | NA | <0.01 | 1.43 | 2.00 | NA | NA | 1.26 | 1.90 |
| Carangidae | *Carangoides* sp. | NA | NA | NA | NA | NA | NA | 2.00 | 2.57 | NA | NA | 1.26 | 2.36 |
| Carangidae | *Megalaspis cordyla* | 0.02 | <0.01 | NA | NA | <0.01 | NA | 1.43 | 2.29 | 1.94 | 2.41 | NA | NA |
| Leiognathidae | *Eubleekeria splendens* | 0.09 | 0.21 | NA | NA | 0.05 | NA | 1.14 | 1.86 | 1.62 | 1.98 | NA | NA |
| Leiognathidae | *Gazza minuta* | 0.22 | 0.03 | 0.07 | <0.01 | 0.01 | 0.02 | 1.14 | 1.71 | 1.94 | 2.26 | 1.26 | 1.70 |
| Leiognathidae | *Nuchequula gerreoides* | 0.04 | 0.02 | 0.27 | 0.06 | 0.01 | 0.02 | 1.14 | 1.86 | 1.94 | 2.26 | 1.26 | 1.70 |
| Leiognathidae | *Secutor hanedai* | NA | NA | <0.01 | <0.01 | NA | <0.01 | NA | 1.86 | NA | NA | 1.41 | 1.81 |
| Lutjanidae | *Lutjanus russelli* | 0.04 | 0.04 | <0.01 | <0.01 | 0.03 | <0.01 | 1.57 | 2.00 | 2.45 | 2.91 | 1.26 | 2.01 |
| Gerreidae | *Gerres macracanthus* | <0.01 | <0.01 | NA | NA | <0.01 | NA | 1.29 | 1.86 | 2.18 | 2.53 | NA | NA |
| Haemulidae | *Pomadasys kaakan* | NA | NA | 0.09 | 0.04 | NA | <0.01 | 2.00 | 2.00 | NA | NA | 1.26 | 2.36 |
| Haemulidae | *Pomadasys maculatus* | NA | NA | <0.01 | <0.01 | NA | <0.01 | 1.86 | 2.00 | NA | NA | 1.26 | 2.24 |
| Polynemidae | *Eleutheronema tetradactylum* | <0.01 | <0.01 | NA | NA | <0.01 | NA | 2.00 | 1.43 | 2.33 | 3.07 | NA | NA |
| Sciaenidae | *Johnius amblycephalus* | 0.09 | 0.08 | 0.13 | 0.07 | 0.03 | 0.01 | 1.29 | 2.00 | 2.62 | 2.92 | 1.26 | 1.80 |
| Sciaenidae | *Pseudosciaena soldado* | 0.48 | 0.38 | 0.04 | 0.09 | 0.22 | <0.01 | 1.86 | 1.71 | 2.18 | 2.87 | 1.26 | 2.24 |
| Sciaenidae | *Otolithes ruber* | 0.65 | 0.65 | <0.01 | <0.01 | 0.33 | <0.01 | 1.43 | 1.43 | 2.18 | 2.61 | 1.26 | 1.90 |
| Sciaenidae | *Pennahia anea* | 0.13 | 0.03 | 0.05 | 0.02 | 0.04 | <0.01 | 1.29 | 1.43 | 2.18 | 2.53 | 1.59 | 2.04 |
| Sciaenidae | *Panna microdon* | 0.04 | 0.01 | NA | NA | 0.01 | NA | 1.29 | 2.00 | 2.62 | 2.92 | NA | NA |
| Mullidae | *Upeneus sulphureus* | <0.01 | <0.01 | 0.07 | 0.05 | <0.01 | <0.01 | 1.14 | 2.00 | 2.18 | 2.46 | 1.26 | 1.70 |
| Mullidae | *Upeneus sundaicus* | NA | NA | 0.25 | 0.28 | NA | 0.02 | 1.29 | 2.00 | NA | NA | 1.41 | 1.91 |
| Drepaneidae | *Drepane punctata* | 0.74 | 0.74 | NA | NA | 0.38 | NA | 1.57 | 1.86 | 2.62 | 3.06 | NA | NA |
| Ephippidae | *Ephippus orbis* | <0.01 | <0.01 | NA | NA | <0.01 | NA | 1.29 | 2.14 | 2.45 | 2.77 | NA | NA |
| Scatophagidae | *Scatophagus argus* | NA | NA | 0.04 | <0.01 | NA | <0.01 | 1.14 | 1.29 | NA | NA | 1.41 | 1.80 |
| Sphyraenidae | *Sphyraena jello* | NA | NA | 0.11 | <0.01 | NA | 0.01 | 2.14 | 2.00 | NA | NA | 1.26 | 2.49 |
| Stromateidae | *Pampus chinensis* | <0.01 | 0.06 | NA | NA | 0.01 | NA | 1.29 | 1.86 | 2.33 | 2.67 | NA | NA |
| Blenniidae | *Petroscirtes* sp. | NA | NA | 0.02 | 0.01 | NA | <0.01 | 1.29 | 2.71 | NA | NA | 1.26 | 1.80 |
| Gobiidae | *Acentrogobius caninus* | NA | NA | 0.05 | 0.01 | NA | 0.01 | 1.43 | 2.00 | NA | NA | 1.26 | 1.90 |
| Siganidae | *Siganus canaliculatus* | NA | NA | 0.23 | 0.25 | NA | 0.02 | 1.14 | 1.86 | NA | NA | 1.41 | 1.81 |
| Siganidae | *Siganus javus* | 0.04 | 0.09 | 0.32 | 0.46 | 0.01 | 0.06 | 1.14 | 1.86 | 2.62 | 2.86 | 1.41 | 1.81 |
| Scombridae | *Scomberomorus commerson* | 0.04 | 0.09 | NA | NA | 0.08 | NA | 2.00 | 1.86 | 2.04 | 2.86 | NA | NA |
| Cynoglossidae | *Cynoglossus arel* | NA | NA | 0.04 | 0.01 | NA | <0.01 | 1.43 | 2.00 | NA | NA | 2.14 | 2.57 |
| Cynoglossidae | *Cynoglossus trulla* | 0.04 | 0.05 | 0.02 | 0.01 | 0.02 | <0.01 | 1.43 | 2.00 | 2.80 | 3.15 | 1.59 | 2.14 |
| Cynoglossidae | *Cynoglossus* sp. 1 | 0.04 | 0.09 | 0.07 | 0.02 | 0.02 | 0.01 | 1.43 | 2.57 | 2.80 | 3.15 | 2.14 | 2.57 |

**Table 3** (*continued*)

| Family | Scientific name | % N (G) | %W (G) | %N (T) | %W (T) | %IRI (G) | %IRI (T) | P | QP | S (G) | V (G) | S (T) | V (T) |
|---|---|---|---|---|---|---|---|---|---|---|---|---|---|
| Cynoglossidae | *Cynoglossus* sp.2 | 0.04 | 0.01 | 0.02 | 0.07 | 0.01 | <0.01 | 1.43 | 2.57 | 2.80 | 3.15 | 1.91 | 2.39 |
| Soleidae | *Brachirus orientalis* | 0.87 | 1.63 | 0.36 | 0.49 | 0.98 | 0.13 | 1.43 | 2.00 | 2.8 | 3.15 | 1.59 | 2.14 |
| Soleidae | *Brachirus harmandi* | 0.09 | 0.11 | 0.09 | 0.02 | 0.03 | <0.01 | 1.29 | 2.00 | 2.8 | 3.08 | 1.91 | 2.3 |
| Soleidae | *Synaptura commersonnii* | <0.01 | <0.01 | NA | NA | <0.01 | NA | 1.43 | 2.14 | 2.8 | 3.15 | NA | NA |
| Monacanthidae | *Paramonacanthus choirocephalus* | NA | NA | 1.22 | 0.16 | NA | 1.60 | 1.29 | 2.00 | NA | NA | 1.41 | 1.91 |
| Tetraodontidae | *Chelonodon* sp. | NA | NA | 0.09 | 0.42 | NA | 0.02 | 1.57 | 2.57 | NA | NA | 1.26 | 2.01 |
| Tetraodontidae | *Lagocephalus lunaris* | <0.01 | <0.01 | 1.67 | 0.36 | <0.01 | 1.06 | 1.57 | 2.00 | 1.94 | 2.5 | 1.41 | 2.11 |
| Tetraodontidae | *Takifugu oblongus* | <0.01 | <0.01 | 2.94 | 18.79 | <0.01 | 3.62 | 1.43 | 2.14 | 1.82 | 2.31 | 1.41 | 2.01 |

**Notes.**

G and T are stood for gillnet and trap, respectively. %N, %W and %IRI are percentages in number, weight and index of relative importance, respectively. The scores from productivity-susceptibility analysis are P = overall productivity score, QP = data quality score for productivity attributes, S= overall susceptibility score and V= total vulnerability score. NA means species was not available in the catches.

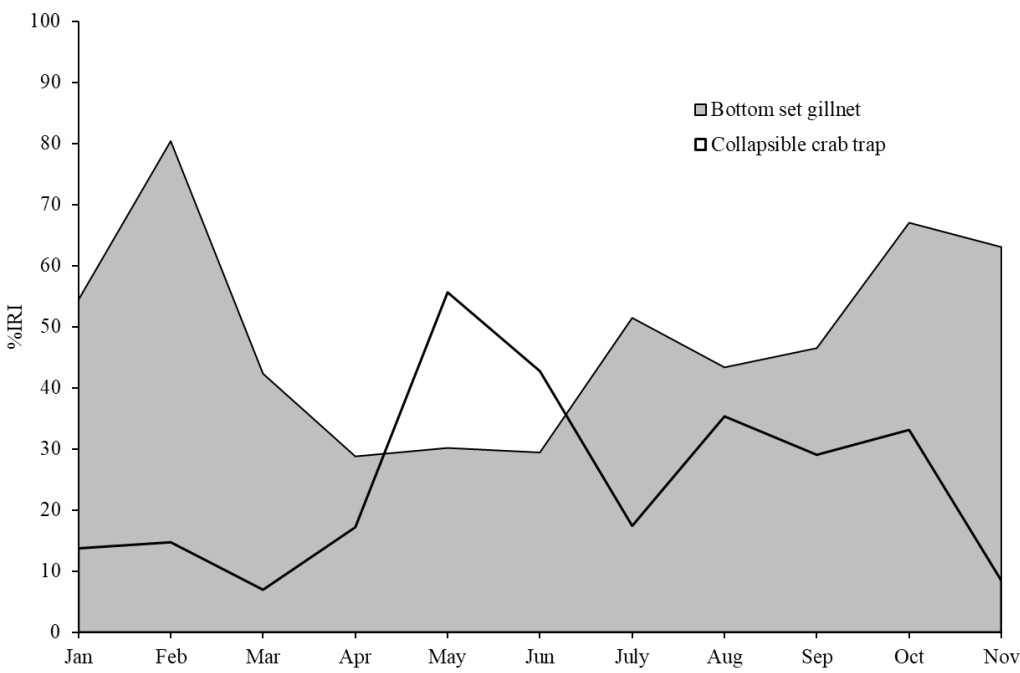

**Figure 3** Index of relative importance of blue swimming crab, as main target species, in bottom-set gillnets and collapsible crab traps in Bandon Bay, Surratthani, Thailand.

itself might place the trap in an inappropriate position, in particular the entrance, and lead to lower catches of all species quantitatively and qualitatively. Gillnets, on the other hand, would still continue to function during the monsoon season due to the length of the nets and no significant difference in catches by different hanging ratios of the nets (*Gray et al., 2005*). The higher number of species captured during summer in both gears, though many were non-target species, could be due in part to the good conditions for fishing operations. Variation in species composition between the monsoon and non-monsoon seasons was also observed in gillnets and traps in the lower and eastern Gulf of Thailand, respectively (*Fazrul et al., 2015*; *Kunsook & Dumrongrojwatthana, 2017*). Fewer fish species in catches during the monsoon could be caused by freshwater discharge to the bay, which forces marine fishes further offshore (*Jutagate et al., 2010*; *Jutagate et al., 2011*).

Using PSA to assess the impacts of fisheries to fish stocks has increased recently, in particular for multi-species fisheries, where information on stock status of non-targeted species is always lacking or limited (*Hordyk & Carruthers, 2018*; *Lin et al., 2020*; *Faruque & Matsuda, 2021*). By screening the high or relatively high-risk species from both gears, through PSA, these species can be then taken into consideration for assessing their stock status, accompanied with the main target species, for further implementing appropriate measures to sustain the fisheries. Although several attributes have been added to PSA recently, such as in extended PSA (*Hordyk & Carruthers, 2018*) and revised PSA (*Grewelle et al., 2021*), we chose to use the standard PSA (*Hordyk & Carruthers, 2018*) in this study since we were able to integrate available attribute data with local knowledge from

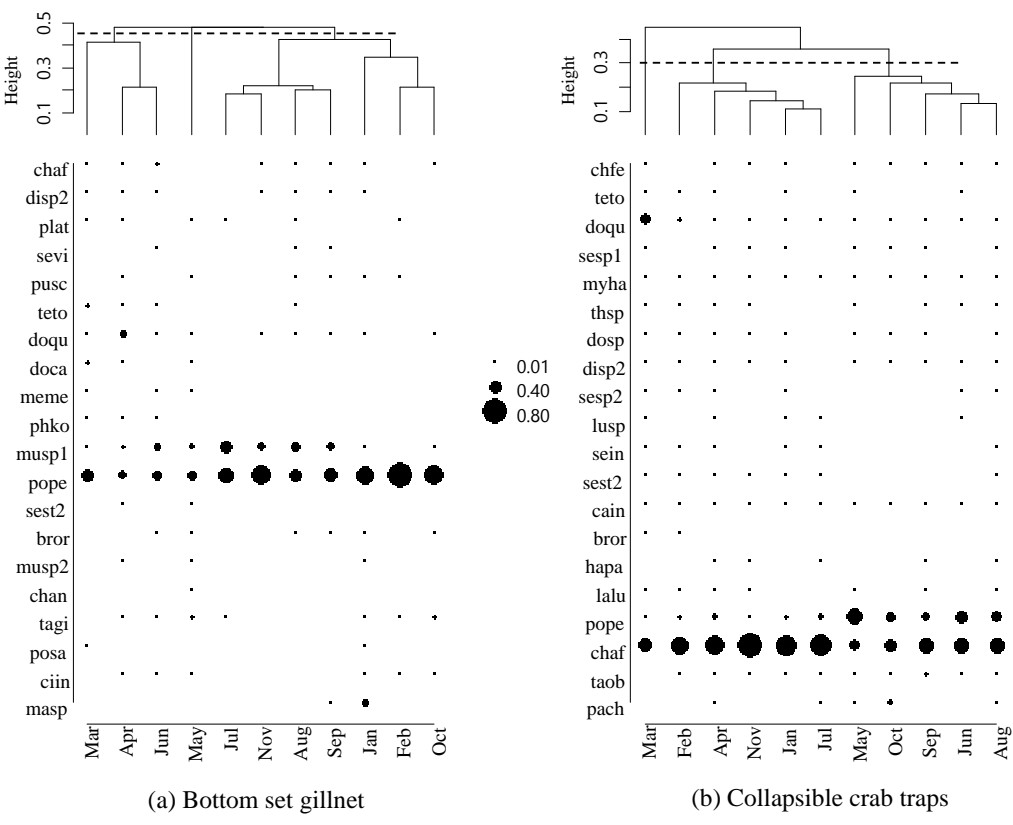

**Figure 4** **Dendrogram cluster by month of sampling of main catches by (A) bottom-set gillnets and (B) collapsible crab traps in Bandon Bay, Surratthani, Thailand.** Abbreviations: chaf, Crabs *Charybdis affinis*; sevi, *Seulocia vittata*; doqu, *Dorippe quadridens*; doca, *Doclea canalifera*; (pope), *Portunus pelagicus*; chan, *Charybdis anisodon*; posa, *Portunus sanguinolentus*; masp, *Macrophthalmus* sp.; chfe, *Charybdis feriata*; myha, *Myomenippe hardwickii*; thsp, *Thalamita spinimana*; dosp, *Doclea* sp.; plat, Bony fishes *Platycephalus* sp.; bror, *Brachirus orientalis*; lalu, *Lagocephalus lunaris*; taob, *Takifugu oblongus*; pach, *Paramonacanthus choirocephalus*; pusc, Gastropods *Pugilina schumacher*; Murex, *Melo melo* (meme); musp1, sp.1; musp2, *Murex* sp.2; sesp1, Cephalopods *Sepia* sp.1; sesp2, *Sepia* sp.2; sein, *Sepiella inermis*; disp2, Hermit crabs *Diogenes* sp.2; ciin, *Clibanarius infraspinatus*, cain, *Clibanarius infraspinatus*; teto, Sea stars *Temnopleurus toreumaticus*; sest2, Sea star 2; phko, Sea cucumber *Phyllophorella kohkutiensis*; tagi, Horseshoe crab *Tachypleus gigas*; lusp, Brittle star: *Luidia* sp.; hapa, Mantis shrimp: *Harpiosquilla harpax*.

fishers. Their knowledge is very crucial for the susceptibility attributes and also useful for identifying important local differences in stock susceptibility to fishing (*Jara, Damiano & Heppell, 2022*). *Robinson, Cinner & Graham (2014)* reported a good understanding and homogenous knowledge of susceptibility to fishing gears displayed by fishers that operate the same fishing gear, have access to the same fishing ground and have similar economic background. Moreover, rank scores of susceptibility generated from documents, by the research team, and by other scientists were identical. For productivity attributes, *Lin et al. (2020)* mentioned that although maximum size and size at 50% maturity may show autocorrelation, they must both be kept in the model since they describe distinctly different biological components of a species' life history. The data quality scores for these attributes of BSC and some other aquatic animals (*e.g.,* mud crab, prawns, sea cucumbers, some
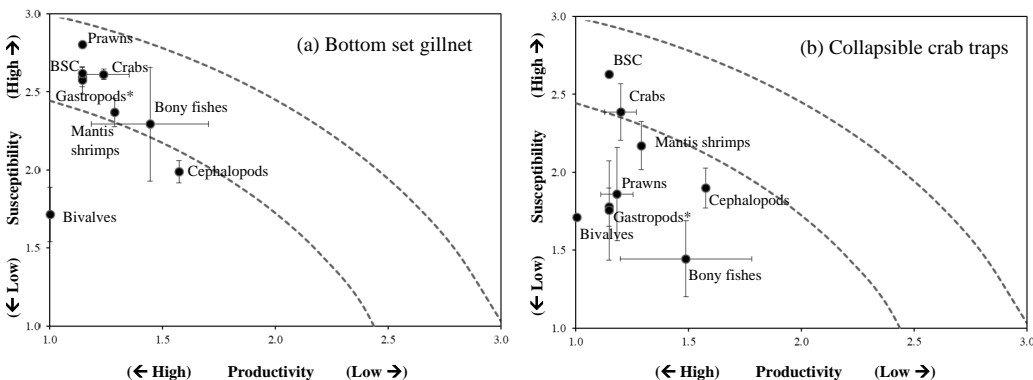

**Figure 5** **Productivity-susceptibility plot for blue swimming crab and other catch-groups by (A) bottom-set gillnets and (B) collapsible crab traps in Bandon Bay, Surratthani, Thailand.** Lines indicate standard deviations of productivity and susceptibility attributes.

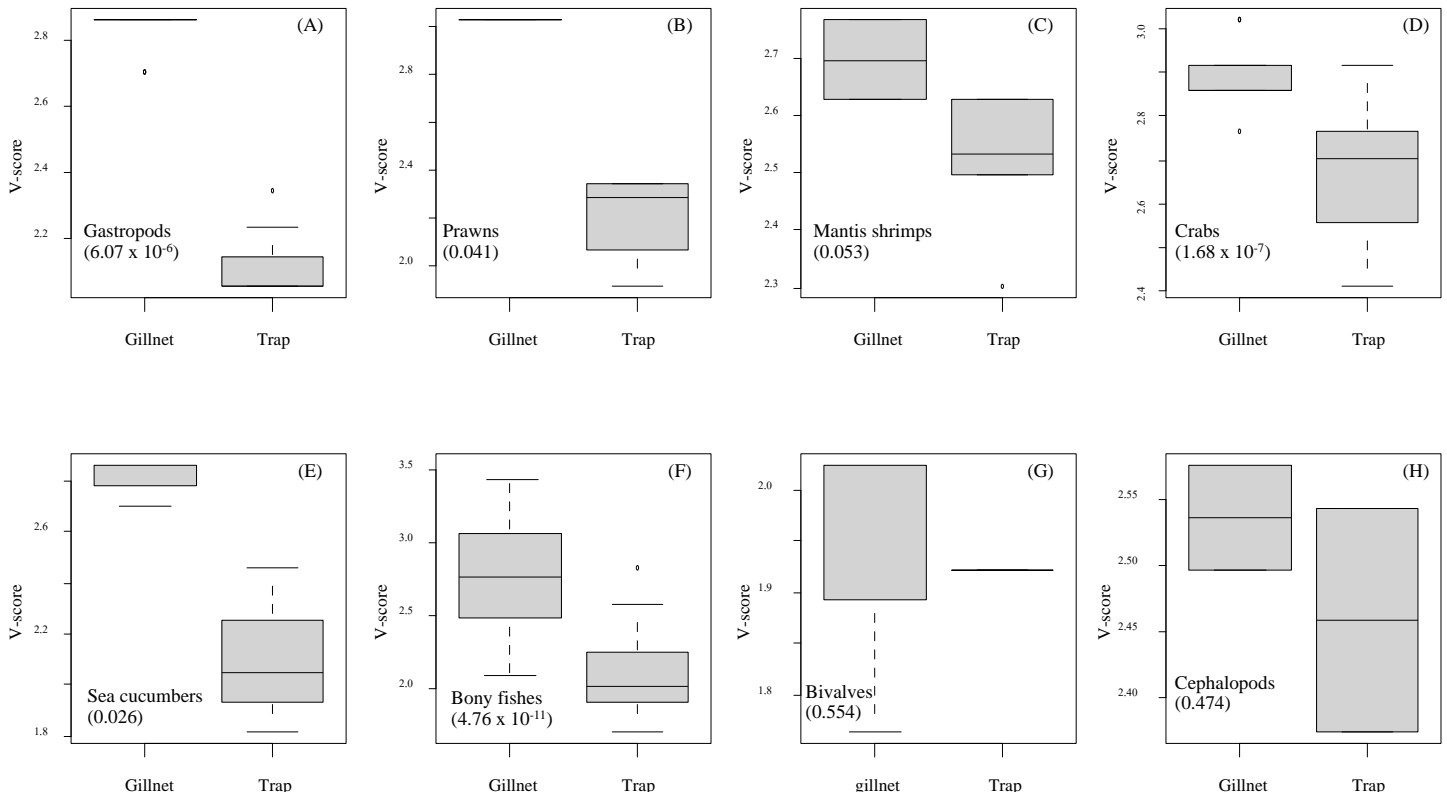

**Figure 6** **(A–H) Box-plots showing the vulnerability scores between two fishing gears for each group of aquatic animals.** Gillnet = bottom-set gillnet and Trap = collapsible crab trap. Number in parentheses is the *P*-value from Mann–Whitney *U* test.

fishes) were available because of their market value, and hence, have received more study. However, as in other tropical marine fisheries, data quality scores were limited for species with little or no market value, including crabs, other aquatic animals and fishes (*Lin et al., 2020*; *Faruque & Matsuda, 2021*).

Gillnets and traps cause considerably lower holistic environmental impacts than active fishing gears (*Uhlmann & Broadhurst, 2015*). Vulnerability of the BSC stock, as the main targeted species, to gillnets and traps in SSF of Bandon Bay was at a moderate level and similar to the BSC stock of Phu Quoc Island, Vietnam (*Ha et al., 2015*). Meanwhile, the stocks of fishes and other aquatic animals in Bandon Bay were more vulnerable to gillnets than traps. This is due to the fact that the discard mortality by gillnets is relatively high, with a reported mean of about 40% across the range of species, and is considerably lower in traps (*Uhlmann & Broadhurst, 2015*). The low to moderate risk found for almost all species is likely due to their potential to recover their stocks, with recovery capacity ranges between 1 and 5 years for most tropical fishes (*Mohamed & Veena, 2016*). *Mohamed et al. (2021)* reported that most of the fish stocks along the coast of India were resilient-yet-vulnerable, and most crustaceans showed high resilience. Higher vulnerability of the two stingrays in this study is due to their life history; like most elasmobranchs, they have low fecundity, exhibit ovoviviparity, and are carnivorous (*Frisk, Miller & Fogarty, 2001*; *Mohamed et al., 2021*). Productivity attributes also make *M. cinereus* and *H. sagor* more vulnerable because of their elongate form with high maximum size and trophic level for the former and low fecundity, late maturity and carnivorous diet of the latter (*Kottelat, 2013*; *Sang, Lam & Hai, 2019*; *Froese & Pauly, 2021*). On the other hand, high risk to soles by gillnets is largely caused by their susceptibility, resulting in either moderate or high risk scores for all attributes.

A mesh size regulation (not less than 2.5 inch) is currently applied to both fishing gears. However, this regulation may less effective for gastropods and crustaceans since they are always entangled in the gillnets (*Fazrul et al., 2015*; *Faruque & Matsuda, 2021*). Other relevant measures to both SSF in Bandon Bay are a spatial closure and efforts at stock enhancement. The goal of the spatial closure is to create fishery refugia, and was established at Sed Island in 2021 (Fig. 1). It is an attempt to restore the stocks of many species in Bandon Bay, because the area is important nursery habitat for a number of fishes and other aquatic animals, including the BSC (*Thongkhao, 2020*). In terms of enhancement, stocking has focused on the BSC through the "crab bank" project to preserve and disperse eggs post capture. The aim is to increase recruitment of BSC, which consequently sustains the gillnet and trap SSF in Bandon Bay.

## CONCLUSIONS

In Bandon Bay, over 100 species of fishes and other aquatic animals were caught in gillnets and traps, confirming the high productivity of this fishing ground and the multi-species nature of the SSF (*Sawusdee, 2010*). Significantly higher %IRI of BSC compared to other species in both gears almost year-round suggest an abundance of BSC and the relative specificity of the gears. The PSA indicated low to moderate risk from BSC fisheries to

the stocks of other species in the catches, although stingrays and eight (8) bony fishes were were near the threshold of high risk from gillnet fishing, implying that both fishing gears are not excessively impactful and are appropriate for use by the SSF of Bandon Bay. Nevertheless, risk may be underestimated by applying PSA, as cautioned by *Grewelle et al. (2021)*, and this should be taken into consideration when implementing the results for fisheries management. Catch monitoring and stock assessment of both targeted and non-targeted species should be regularly conducted (*Lin et al., 2020*; *Prince et al., 2020*). Impacts from other stressors (*e.g.*, climate change, sea ranching and land uses) should be taken into consideration to sustain the fishery resources and the fisheries in Bandon Bay.

## ACKNOWLEDGEMENTS

We are grateful to the 60 fishers and 15 fishery scientists for their involvement and sharing their knowledge in our PSA study.

### Funding
This study was financially supported by the Agricultural Research Development Agency (Public organization) (project number: PRP6005010660) and the Thai Frozen Foods Association under the Fisheries Improvement Project of the blue swimming crab fisheries at Suratthani Province, Thailand. The funders had no role in study design, data collection and analysis, decision to publish, or preparation of the manuscript.

### Grant Disclosures
The following grant information was disclosed by the authors:
Agricultural Research Development Agency: PRP6005010660.
Thai Frozen Foods Association.

### Competing Interests
The authors declare there are no competing interests.

### Author Contributions
- Tuantong Jutagate conceived and designed the experiments, performed the experiments, analyzed the data, prepared figures and/or tables, authored or reviewed drafts of the article, and approved the final draft.
- Amonsak Sawusdee conceived and designed the experiments, performed the experiments, analyzed the data, prepared figures and/or tables, authored or reviewed drafts of the article, and approved the final draft.

### Animal Ethics
The following information was supplied relating to ethical approvals (i.e., approving body and any reference numbers):

The Institute of animals for scientific purposes development granted approval for this research (U1-04118-2559).

## Field Study Permissions

The following information was supplied relating to field study approvals (i.e., approving body and any reference numbers):

Field experiments were approved by the Agricultural Research development agency (Public organization) (project number: PRP6005010660).

## Data Availability

Raw data is available in the Supplementary Files.

## Supplemental Information

Supplemental information for this article can be found online at http://dx.doi.org/10.7717/peerj.13878#supplemental-information.

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
