# Peer review of "Catch composition and risk assessment of two fishing gears used in small-scale fisheries of Bandon Bay, the Gulf of Thailand"

_PeerJ, doi:10.7717/peerj.13878_

## Round 0.1 · original submission · Minor Revisions

The research design, results and discussion of the manuscript are sound. However, I concur with the reviewers that the English would need some proofreading. I look forward to your revision.

Reviewer 1 ·

Basic reporting

1. Inconsistencies of the use between British and American English
2. Grammar is acceptable, but can be fine tuned
3. Inconsistencies with the use of "and" and "&" in in-text citation
4. Some references are missing in the in text citation

Experimental design

1. Well structured and conceptualised.
2. Hypothesis are in line with objectives and methodology

Validity of the findings

1. Impactful, as certain fishery findings tends to be overlooked if its not commercial. It is also good to see which gears have potential impact towards the environment or catch. Please refer to my word document comments to polish certain aspects of the content.

Additional comments

All well, no comments. Previous reviewer's comments have also been addressed

Annotated reviews are not available for download in order to protect the identity of reviewers who chose to remain anonymous.

·

Basic reporting

In general, the research is quite good, the sampling location is representative of the research location and is carried out in a relatively long time. Conditions of tropical waters have shown seasonal variations throughout the year.
The effect of fishing gear on fish stocks, it is necessary to look at the selectivity of fishing gear, including the selectivity of species and sizes. When referring to CCRF, fishing gear must be selective, selective on the type and size of fish.
In referring to the sustainability of fishery resources, the sustainability of gill nets and traps is still in question. Why, because too many fish species are caught, in addition, the size of the main fish target is also not clear, moreover the bycatches.
The main target of crabs, but there is no information on the size of crab’s worth catching. So, the sustainability of crabs is still questionable, especially other fish which are by-catch.
SDG 14 sustainable development – refers to sustainable fisheries
In line 19 it is mentioned analyzing the target species and bycatch but it does not describe in detail the main catch and bycatch of the two fishing gears. The main catch also needs to be evaluated whether it is worth catching or not. The same goes for bycatch. The size of the fish caught is very important in considering the sustainability of fishery resources.

Experimental design

In lines 29-31 it is stated that the impact of the 2 fishing gears on the resource is still moderate, why not at a high level. The facts show that the number of species caught is large and it is likely that the size of the fish will also vary greatly. From the point of view of selectivity and sustainability, the two fishing gears tend to interfere with the sustainability of fish resources at the research site.
Lines 104-105 refer to sustainable fisheries, but in practice and discussion it does not show the sustainability of the fishing gear used. Why?
Line 153 refers to table 1 of FAO, but there is no data regarding this indicator. Why? The existing data should be processed referring to the indicators in table 1. How the data collected should be explain in detail. Line 157-158, how FGD decided the size of fish maturity, gear selectivity etc.? need more information.
Observations made throughout the year should provide an overview of the size distribution of fish and spawning times so that management strategies can be used.

Validity of the findings

In general, the findings are good but it would be even better if the data on the size of the main target fish and bycatch were added. Why is this important, referring to SDG 14. In line 185, how can gastropods be caught in a gill net? Lines 185-189 describe the highest percentage of 5 species based on number but on line 190 do not use the same method on composition based on fish weight, it should be consistent.
In the discussion the author should explain the research findings and why the results are like that. One of the critical points that need to be explained is the impact of fishing gear on the main target related to the size of fish suitable for catching, referring to sustainable fishing technology. In addition, it is necessary to highlight the selectivity of fishing gear because there are too many by-catch.
In the discussion section it is necessary to discuss why too many species of fish are caught in these two fishing gears and how to reduce them. According to the code of ethics for responsible fisheries, fishing gear must be selective.
Line 321 The conclusion section refers to the purpose of the study without any discussion and reference.
Conclusion should be in line with abstract, line 33-34.

Additional comments

In general, the research is quite good but it is necessary to emphasize the discussion related to sustainable fisheries as referred to in SDG14. In the methodology several important points need to be clarified especially in relation to the FAO indicators in table 1 and how the data were obtained. Further refined on the results and discussion. The discussion needs to explain more about research findings related to the conditions of the research location, time, habitat and other conditions. Furthermore, it is compared with the findings of other researchers. At the end, the conclusion should no longer be discussed and must be in line with the abstract.

Reviewer 3 ·

Basic reporting

The whole manuscript needs english proofreading. There are various instances of grammatical errors that would obscure the intended meaning of the sentences. Also, in the Introduction section, it would be great if the authors could show to the readers the impact of gillnets and traps in other small-scale fisheries (Line 97-98). Figures are well developed. Results supported the hypotheses, where the authors revealed the catch composition and risk assessment of using two gears at GoT.

Experimental design

The experimental design is valid and sound. At least 1 year of sampling was conducted and it fills a quite important knowledge gap, serving baseline data for future fishery studies and management. Method section is clearly described.

Validity of the findings

Findings were analyzed with the appropriate data analysis.

---

## Round 0.2 · accepted · Accept

Thank you for following through with all the comments and suggestions from me and the reviewers.